# Graphene: Hexagonal Boron Nitride Composite Films with Low-Resistance for Flexible Electronics

**DOI:** 10.3390/nano12101703

**Published:** 2022-05-17

**Authors:** Irina V. Antonova, Marina B. Shavelkina, Artem I. Ivanov, Dmitriy A. Poteryaev, Nadezhda A. Nebogatikova, Anna A. Buzmakova, Regina A. Soots, Vladimir A. Katarzhis

**Affiliations:** 1Rzhanov Institute of Semiconductor Physics SB RAS, 13 Lavrentiev Aven., Novosibirsk 630090, Russia; art.iv.il@mail.ru (A.I.I.); poteryayevd@inbox.ru (D.A.P.); nadonebo@gmail.com (N.A.N.); soots@isp.nsc.ru (R.A.S.); 2Department of Semiconductor Devices and Microelectronics, Novosibirsk State Technical University, 20 K. Marx Str., Novosibirsk 630073, Russia; kiraromanova011@gmail.com; 3Joint Institute for High Temperatures RAS, Izhorskaya Str. 13 Bd.2, Moscow 125412, Russia; mshavelkina@gmail.com (M.B.S.); korg983@yandex.ru (V.A.K.)

**Keywords:** DC plasma synthesis, composite nanoparticle graphene:h-BN:PEDOT:PSS, structure, electric properties, self-organization effect, inkjet 2D printing technologies

## Abstract

The structure and electric properties of hexagonal boron nitride (h-BN):graphene composite with additives of the conductive polymer PEDOT:PSS and ethylene glycol were examined. The graphene and h-BN flakes synthesized in plasma with nanometer sizes were used for experiments. It was found that the addition of more than 10^−3^ mass% of PEDOT:PSS to the graphene suspension or h-BN:graphene composite in combination with ethylene glycol leads to a strong decrease (4–5 orders of magnitude, in our case) in the resistance of the films created from these suspensions. This is caused by an increase in the conductivity of PEDOT:PSS due to the interaction with ethylene glycol and synergetic effect on the composite properties of h-BN:graphene films. The addition of PEDOT:PSS to the h-BN:graphene composite leads to the correction of the bonds between nanoparticles and a weak change in the resistance under the tensile strain caused by the sample bending. A more pronounced flexibility of the composite films with tree components is demonstrated. The self-organization effects for graphene flakes and polar h-BN flakes lead to the formation of micrometer sized plates in drops and uniform-in-size nanoparticles in inks. The ratio of the components in the composite was found for the observed strong hysteresis and a negative differential resistance. Generally, PEDOT:PSS and ethylene glycol composite films are promising for their application as electrodes or active elements for logic and signal processing.

## 1. Introduction

The composite materials are formed by mingling at least two or more materials having different properties, resulting in the unique output characteristics of the composite material [1,2]. Additional benefits of composite materials are the high customization in design because the composite materials can be formed into complex shapes or create unusual composite nanoparticles. With the development and advancement of next-generation composite materials, they have been widely used not only in the aerospace industry but also in the mechanical, construction, biomedical, automobile and optoelectronic industries and in electronics [3,4,5]. Two-dimensional-layered materials (graphene, transition metal dichalcogenides, hexagonal boron nitride, etc.) have been used as attractive platforms due to their mechanical strength and flexibility in electric and optical properties [3,4,5,6,7]. The development of these composites based on 2D nanomaterials brings great achievements with novel applications that have been unknown before. 

The strategies of composite synthesis, characterization techniques and the mechanism of interaction between components are thoroughly discussed and examined since they are pivotal to understanding the properties of the composites (composite nanoparticle morphology, size, structure and dimensionality of composites, etc.). The possibility to combine different 2D materials in one layer allows one to additionally manage the electron and optical properties of the resulting composite. The widely used method of mechanical transfer of individual 2D crystals provides the quality of fabricated heterostructures and interfaces, but it is not scalable [8,9,10]. The new approach to creating heterostructures using two-dimensional (2D) atomic flakes, including the utilization of the printed technologies, led to exciting physical phenomena and the development of novel devices [11,12,13]. The composite layers created from the materials used for electronics modify their properties and expand their possible applications, including flexible electronics [14,15,16]. Here, we show that such composite layers can be used for low-cost and scalable printing technologies of the device fabrication.

At present, h-BN:graphene composites are widely studied to control the thermal conductivity of the composite material [17,18]. The thermal conducting properties of h-BN in composites minimize the thermal interface resistance and increase the phonon conduction. One of the significant h-BN challenges is the lack of scalable production with tunable configurations and controllable quality, including the development of large-area and high-quality growth or synthesis processes of h-BN flakes with good reliability and repeatability. Innovations in growing customized 2D h-BN and its nanoflakes for specific application requirements are of high demand to develop next-generation integrated devices. The synthesis with the use of DC plasma torch is a perspective for an industrial production.

An increase in graphene synthesis temperature leads to improving its quality. In our studies, we used materials synthesized under extreme plasma conditions. Plasma methods make it possible to control the synthesis process and control the geometry of the synthesis products [19]. In the present study, two opposing top-down for graphene and bottom-up for h-BN approaches were implemented. The resulting uniformity in size, thickness and properties of h-BN and graphene flakes were used for the creation of composite particles and layers. The h-BN:graphene composite with additives of the conductive polymer PEDOT:PSS allows us to create composite inks for the inkjet 2D printing of low-resistance layers (sheet resistance 20–70 Ohm/sq) that demonstrate their negative differential resistance or memristor characteristics. The self-organization effects for graphene flakes and polar h-BN flakes lead to the formation of micrometer-sized plates in drops and uniform-in-size nanoparticles in inks.

## 2. Experimental

### Plasma Synthesis of Graphene Flakes and h-BN Nanosheets

To synthesize graphene, we applied the plasma-chemical approach based on a 40 kW DC plasma torch with the expanding output electrode channel and the plasma jet vortex stabilization [19]. The design feature of the plasma torch (expanding output electrode channel) and the tangential input make it possible to obtain stable plasma jets within a wide range of parametric studies. A more detailed description of the installation is given in [19,20]. In contrast to the process of graphene synthesis, where the bottom-up approach was realized, the synthesis process of graphene-like BN flakes was carried out according to the top-down principle. As a precursor, we used h-BN powder, which is known as a component of structural ceramics.

The product of the plasma–chemical synthesis is graphene powder (the so-called dry powder) with a typical bulk density of 0.1 mg/cm^3^. As shown in [21], the synthesis of graphene flakes in helium plasma leads to the formation of conductive graphene. For the study, we used two types of graphene flakes named G-1 and G-2. In the first case, the 1–3 monolayer flakes sized 100–150 nm are predominant (G-1, Figure 1a). The statement about low thickness of graphene flakes is based on Raman measurements (especially the structure of the line 2D), X-ray structural data and the high-resolution electron microscopy data [20,21,22]. There are the unique parameters, especially in comparison with any graphite exfoliation methods (with typical thickness of 2–5 nm and 5–10 monolayers, e.g., [11,23,24]). The second type of graphene flake has a lower size (<100 nm) and a higher thickness of 1–6 monolayers (G-2, see Figure 1b). The sheet resistance of graphene films created from both suspensions was equal to 300–400 kΩ/sq for a thickness ~1 μm and about 1000 kΩ/sq for 100–200 nm. Such a relatively high resistance is caused by the hydrogen incorporation in graphene layers during its synthesis in plasma. The annealing at 200 °C for 2 h in an inert atmosphere leads to a reduction of the resistance to 4 kΩ/sq due to the escape of hydrogen. The graphene flakes used for composites were not subjected to annealing.

We synthesized the graphene-like structure of hexagonal boron nitride in a plasma chemical reactor with a DC plasma torch. The stable operation of the used plasma torch is ensured by using inert helium as a plasma-forming gas in the pressure range of 500–700 Torr to the atmospheric pressure. Therefore, we used helium both for the synthesis of graphene and for producing boron nitride particles. A series of experiments established that nanosized boron nitride particles (30–60 nm size, see Figure 1c) are formed during the decomposition of the initial h-BN powder at the pressure range 710 Torr. Films created from h-BN flakes have good dielectric properties.

The AFM images and Raman spectra for h-BN and graphene flakes are given in Figure 2. The Raman spectra of graphene and h-BN film contain the standard peaks typical of hexagonal boron nitride (peak 1370 cm^−1^ [5]) and graphene or few-layer graphene: Defect-induced D (1350 cm^−1^), first-order C-C scattering G (1579 cm^−1^) and second-order band 2D (2692 cm^−1^) [25,26]. The 2D-line structure for graphene demonstrates that the synthesized flakes in the film, to a large extent, exhibit their individual properties of 1–3 monolayer flakes [27]. The ratio of line intensities ID/IG < 1 for the graphene film indicates a relatively low defect concentration in graphene flakes. The low ID/IG ratio value is connected with the defect passivation by hydrogen during the graphene synthesis in plasma.

The composite suspension was created from the graphene suspension of 2 mg/mL and BN suspension of 2 mg/mL and then by varying the BN:G suspension content in the range 1:1 to 1:20. The liquid component of all suspensions consists of ethanol (70%) and water (30%). The stirring of suspensions with the use of a shaker leads to the formation of a homogeneous solution.

The conductive polymer PEDOT:PSS (1.1 mass% water solution, high conductive ALDRICH, 739332) was also used for the creation of the composite with graphene. The films created from PEDOT:PSS by thick drops ~1 μm have a resistance of about 6–10 kΩ/sq. For the creation of ink, more additives were created. First of all, there was ethylene glycol (molecular weight 62 a.e.m. and standard and well-known parameters, see, for instance, [28]) with ~30% content from the ink volume. This additive provides the ink viscosity for the inkjet 2D printing, relatively low surface tension and stability of ink properties with temperature variation.

## 3. Experimental Results and Discussion

### 3.1. Morphology and Electric Properties of G:PEDOT:PSS and BN:G:PEDOT:PSS Composites

The effect of PEDOT:PSS polymer addition and PEDOT:PSS plus ethylene glycol on the graphene film resistance is presented in Figure 3. We used ultralow additions to graphene (10^−4^–10^−2^ mass%). The scattering in resistance values in Figure 3 is connected with the variation in the film thickness. The films are created by dropping and the film thickness is varied at different points 2–3 times, which leads to significant changes in resistance. One can see that the addition of PEDOT:PSS leads to a decrease in resistance up to the resistance of pure PEDOT:PSS. This transition starts at the 10^−4^ mass% of PEDOT:PSS for G-1 and at the 2 × 10^−3^ mass% for G-2. Thus, for small graphene flakes, one needs a higher content of PEDOT:PSS to decrease the composite resistance. The additive of the ethylene glycol leads to a strong decrease in the composite resistance. It is well known that PEDOT, which is insoluble in most solvents, can be dispersed in water by using poly(styrene sulfonate) (PSS) as a counter ion, which also serves as an excellent oxidizing agent and charge compensator [29,30]. On the other hand, the PEDOT:PSS addition to the solvent shows somewhat lower conductivities, compared to pure PEDOT. However, the excess PSS is necessary for the dispersion in water. As a result, the addition of solvents, such as ethylene glycol (EG), glycerol and dimethyl sulfoxide, increases the PEDOT:PSS conductivity by up to two or three orders of magnitude [31,32]. The combined results of improved conductivity, reduced thickness and constant transmittance, therefore, indicate that insulating PSS is removed from PEDOT:PSS films during the solvent post-treatment. So, the removal of PSS from the PEDOT:PSS polymer chains is the prime reason for the increased conductivity observed in our experiments.

The films and particles formed in different composites due to the self-organization process are demonstrated in Figure 4 and Figure 5. Particles or plates with different thicknesses are observed (Figure 4a,b and Figure 5). The bright network presented in Figure 4c corresponds to PEDOT:PSS which connects the composite plates. The Appendix A contains a few movies that display the clustering processes in drops of different composites. These movies demonstrated that, in the case of BN:G and BN:G:PEDOT:PSS composites, the dense clusters are effectively formed. We suggested that polar particles of h-BN stimulate the connection of graphene flakes in clusters. PEDOT:PSS plays the role of an additional connecting element (see, for instance, Figure 4b). The sizes of pristine graphene and h-BN flakes are small (from 20 to 150 nm), but plates have sizes of a few micrometers. An interesting structure of relatively thick plates consisting of a few sublayers is given in Figure 5. This large and relatively thick plate is the maximum variant of the clustering process in the used suspensions. The organic additives in the suspension provide a reliable connection between the flakes.

We analyzed the electric properties of the h-BN:G composite and BN:G:PEDOT:PSS composite films. The PEDOT:PSS content was chosen from Figure 3 and, in this case, it was the same and equal to 10^−3^ mass%. In all cases, the graphene content was higher than the percolation threshold for 2D materials (0.1–0.2% [27]). The film resistances versus the composition of the suspension used to obtain composite films are also shown in Figure 6a. This resistance strongly follows the variations in the composite nanoparticle morphology. 

The resistances of the h-BN:G:PEDOT:PSS composite films with G-1 are decreased by four to five orders of magnitudes down to 60–70 or 20 Ohm/sq. In the case of G-2, the resistance is decreased down to values of 20–35 Ohm/sq.

One current-voltage characteristic of the BN:G:PEDOT:PSS composite films with BN:G ratio 1:1 is shown in Figure 6b. Moreover, the characteristics with a negative differential resistance are observed for this composite. The non-linear characteristics in the BN:G:PEDOT:PSS composite films were observed for BN:G ratio 1:1 < X ≤ 1:4. This finding means that graphene flakes are capsulated with h-BN when tunable potential barriers of h-BN create the condition for accumulation of the charge on graphene flakes and formation of low-resistance current paths. As a result, these multibarrier systems demonstrate the hysteresis in I-V curves and the negative differential resistance. For a higher BN:G ratio, the presence of h-BN flakes does not affect the electric properties, and the current-voltage (I-V) characteristics become linear. In the case of the BN:G composite, the I-V characteristics become non-linear with a large hysteresis for the BN:G ratio 1:4 < X ≤ 1:10. Thus, the addition of PEDOT:PSS and ethylene glycol shift the non-linear I-V characteristics to the lower graphene content in composites (1:1 < X < 1:4). Moreover, the I-V characteristics demonstrate the N- shaped negative differential resistance, which is required for some applications in logic and signal processing devices.

### 3.2. Properties of 2D Printed Structures from h-BN:G:PEDOT:PSS Composites

Formation of the composite particles for 2D printing is based on using ultrasonic treatment to limit the clustering process discussed above. This combination leads to the formation of relatively small composite particles. Moreover, the ink composite is more complicated. Ethylene glycol provides the stabilization of the inks and prevents the further clustering. Nevertheless, after the first fast stage of clustering, the second slow stage also occurs. The typical time of this second stage is ~10 days when weak connected clusters of composite particles are formed and the repeated ultrasonic treatment in combination with ink filtration using track membranes before printing allows ones to restore the individual particles in the inks.

In Figure 7 are the AFM images of the printed films and individual clusters formed from composite suspension h-BN:G:PEDOT:PSS (10^−3^ mass%) with the addition of ethylene glycol (30 vol.%) for h-BN:G relation 1:1. We made the structures with two types of layers using 30 and 60 printed passes. The thickness of the layers for G-1 and G-2 in the case of 30 passes was 220–250 nm and 160–200 nm, respectively. First of all, it is worth mentioning that, despite the different sizes of the pristine graphene flakes G-1 and G-2 used to create composites, composite particles are found to be very similar in shape and slightly larger for G-1. It is suggested that PEDOT:PSS take part in the connection of the particles; it can be directly observed between the particles in the top-left part of Figure 7a.

The sheet resistances of the printed layer measured with the use of the four-probe equipment are given in Table 1. The resistance is higher than that obtained by drops because of lower values of the printed film thickness. The current-voltage characteristics measured for printed layers are linear and are not practically changed under tensile strain due to bending.

Testing the composite film flexibility was made on dropped and printed layers with the use of bending (tensile strain). The strain value ε was estimated from equation ε = (d + t)/2r, where d is the PET substrate thickness, t is the film thickness and r is the bending radius. Figure 8a shows the change of resistance for the two printed structures (straight line and wriggle line printed with 30 passes). Despite relatively high h-BN, content ΔR is very weak (3–5%) and that for the wriggle line in both cases for G-1 and G-2 ΔR, as expected, is lower than the change of resistance for the straight line. The change of resistance for the film created by drops from composites with different relations h-BN:G (without PEDOT:PSS) is demonstrated in Figure 8b. It is seen that the increase in the h-BN content leads to a strong increase in the layer resistance. The peak observed in Figure 8b was well reproduced in repeated measurements. The origin of the peak is not known now. It requires further study. We can suggest reorganization of current paths at a certain strain which leads to some decrease in resistance.

The data given in Figure 8a correspond to the maximum used relation h-BN:G = 1:1. One can compare the change of resistance in (a) and (b). Thus, using the ultralow addition of the PEDOT:PSS ~10^−3^ mass% we created a more flexible material based on the h-BN:G composite.

Movies about the h-BN suspension, graphene suspension and its composites drying can be found in the Appendix A. Based on these movies, we can state that h-BN forms relatively small clusters. Graphene flakes create clusters of different sizes, but these clusters are loose. The h-BN:G suspension demonstrates the formation of dense clusters; Figure 4 and Figure 5 show the self-organization effects, which lead to the creation of large (few micrometers) planes in composite films. In the case of printing when the drop size is relatively low, for the composite suspension, we observed particles with similar sizes for both graphene flakes of G-1 and G-2 (see also Appendix A). Appendix A clearly shows the presence of h-BN and G in cluster. The addition of the ultralow portion of PEDOT:PSS does not noticeably influence the drying and cluster formation processes.

## 4. Discussion

Concerning comparison with known composite systems, it worth mentioning that observation of either non-linear I-V characteristics with the negative differential resistance, or memristive swithings, is typical for different composite films and structures [33,34,35]. The origin of these effects is based on formation of tunable potential barriers in composite structures. Different technological findings which provide the possibility to precisely manage with the barrier thickness are profoundly impactful concerning the development of new approaches of composite-based devices. For instance, using composite core-shell nanoparticles, one can control the electrical properties of the created structures and influence their flexibility by means of interparticle connections [22,36]. In our case, the well reproducible dimensions of the elements from which we form composite particles, self-organization processes and synergetic effects associated with the additives used and interlayer (interparticle) interactions make it possible for us to design composite blocks with controlled properties for 2D printing technology.

The interlayer adhesion in the graphene/h-BN multilayered films is of significant importance for their structural and electric properties. It was found that the h-BN/G bilayer deposited on SiO_2_ significantly enlarged (30%) the critical load of the G/SiO_2_ interface failure, compared to that of the graphene monolayer [37]. On the other hand, Leven et al. [38] predicted by a simulation study that a low friction state can exist between the interface of boron nitride and graphene layers in their heterostructures. In another simulation study, Mandelli et al. [39] observed that, for small contact sizes, the heterostructures of graphene and h-BN can also exhibit low friction behavior at heterogeneous junctions during sliding. Low friction behavior is expected in the presence of an organic component in a composite [40]. These factors explain that the conjugate influence of h-BN and graphene nanoparticles can result in different synergistic effects for h-BN/graphene pairs in experimental studies.

## 5. Experimental Techniques for the Samples Characterization

To study the morphology and local properties with a high spatial resolution, we employed Atomic Force Microscopy (AFM) and Raman spectroscopy. The Raman spectra were recorded on a Horiba Jobin Yvon T64000 spectrometer with a 1024-pixel LN/CCD detector and an Ar+ 514.5 nm laser under ambient conditions. A Solver PRO NT-MDT scanning microscope was employed for taking AFM images from the surface of the examined films or individual nanoparticles and for evaluating the sample thicknesses. The measurements were carried out in contact and semicontact modes. The surface morphology of the created films was also studied using a JEOL JSM-7800F (JEOL Ltd., Akishima, Tokyo, Japan) scanning electron microscope (SEM). Then we studied the layer resistance of the fabricated films using four-probe JANDEL equipment and the HM21 Test Unit at room temperature (Jandel Engineering Limited, Limslade, UK). The current-voltage characteristics were also measured with a Keithley picoamperemeter (model 6485) on the samples furnished with two contacts prepared from the silver alloy.

In the fabrication of crossbar structures, the suspensions were applied onto a PET substrate using a Dimatix FUJIFILM DMP-2831 jet printer. Silver contacts were printed with 5-nm particles. After printing five layers, an Ag line 400 nm thick at its centerline was obtained. The sheet resistance was 2.5–3 Ohm sq^−1^.

## 6. Conclusions

The structure, flexibility and electric properties of h-BN:graphene films, with the additives of conductive polymer PEDOT:PSS and ethylene glycol, were examined. The graphene and h-BN flakes synthesized in plasma with nanometer sizes were used for the creation of composites with different contents. It was found that the addition of more than 10^−3^ mass% of PEDOT:PSS, in combination with ethylene glycol (20–30% of inks for the 2D printing technology), to the graphene suspension or h-BN:graphene structure leads to a strong decrease (four to five orders of magnitude in our case) in the resistance of the films created from these suspensions. This is caused by an increase in the PEDOT:PSS conductivity and its synergetic interaction with graphene and h-BN. On the other hand, the self-organization effects for graphene flakes and polar h-BN nanoparticles were revealed, and that leads to the formation of few-micrometer composite plates in the drops and uniform size of the composite nanoparticles in inks and printed layers. Moreover, using small graphene G-2 leads to stronger self-organization effects. One more attractive property of the composite used is the fact that the addition of the PEDOT:PSS to the h-BN:graphene composite leads to the correction of the bonds between nanoparticles and a weak change of the resistance change under the tensile strain caused by bending. Composite films h-BN:graphene:PEDOT:PSS become flexible despite the presence of h-BN flakes.

## Figures and Tables

**Figure 1 nanomaterials-12-01703-f001:**
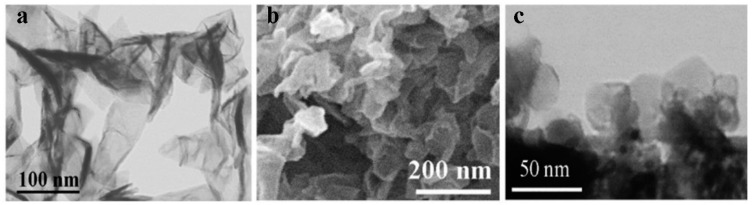
SEM images of pristine flakes G-1 (**a**), G-2 (**b**) and BN (**c**) used for the creation of the composite, respectively.

**Figure 2 nanomaterials-12-01703-f002:**
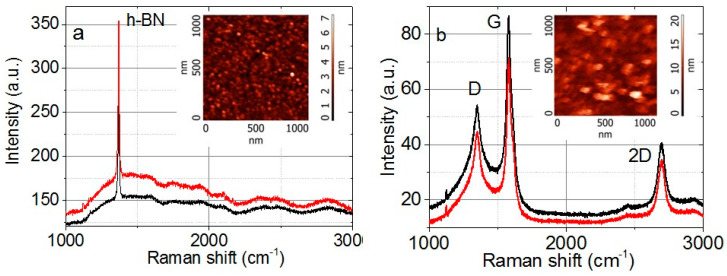
Raman spectra for (**a**) the h-BN powder and (**b**) graphene flakes G-1 measured in two different points on the surface. In the inserts are the AFM images for the h-BN powder and graphene flakes.

**Figure 3 nanomaterials-12-01703-f003:**
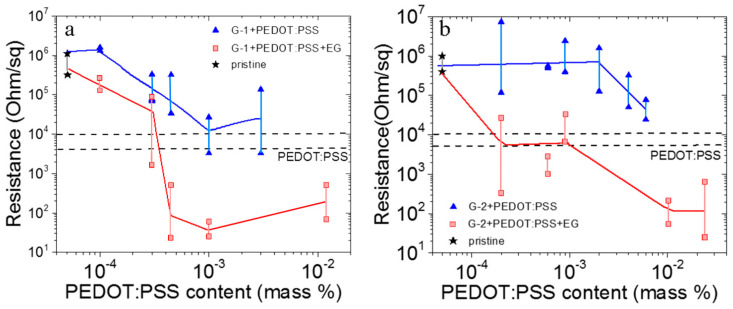
Sheet resistance for composite graphene with PEDOT:PSS (blue points) and graphene, PEDOT:PSS and ethylene glycol (30 vol.%) (red points). Dash lines correspond to PEDOT:PSS films with different thicknesses. (**a**) G-1 (**b**) G-2. Black stars give the resistance values for pure graphene G-1 and G-2. Fluctuation in the sheet resistance connected with a strong (2–3 times) variation of the film thickness created by drops.

**Figure 4 nanomaterials-12-01703-f004:**
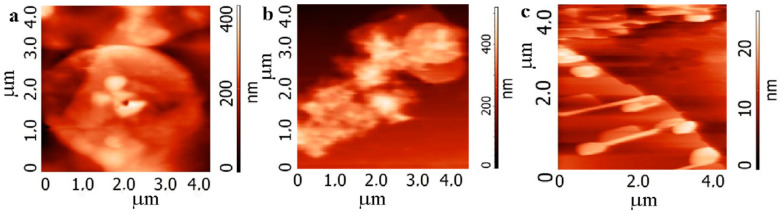
AFM images of thin (100–200 nm) films of the BN:G:PEDOT:PSS composite with the ethylene glycol additive or composite nanoparticles. (**a**,**b**) G-1 with the variation of graphene content BN:G (**a**) 1:4, (**b**) 1:6, (**c**) G-2 content BN:G 1:1.

**Figure 5 nanomaterials-12-01703-f005:**
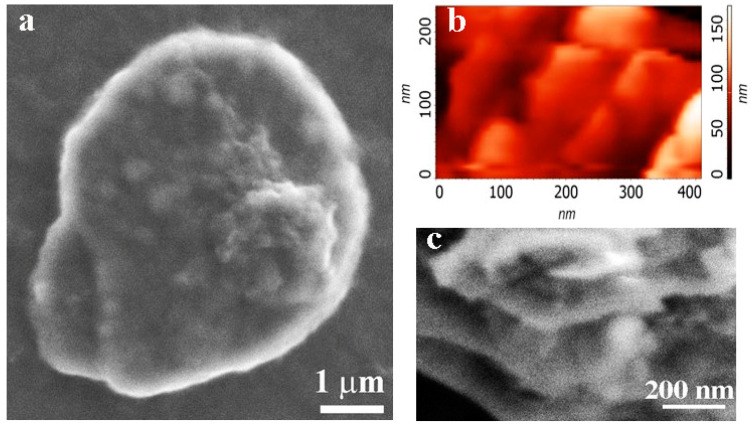
Plate formed due to the self-organization process for graphene G-2 in the film created by the drop from the BN:G:PEDOT:PSS suspension with content BN:G 1:1. (**a**) SEM image of the cluster, (**b**,**c**) AFM and SEM images of the cluster edge, respectively.

**Figure 6 nanomaterials-12-01703-f006:**
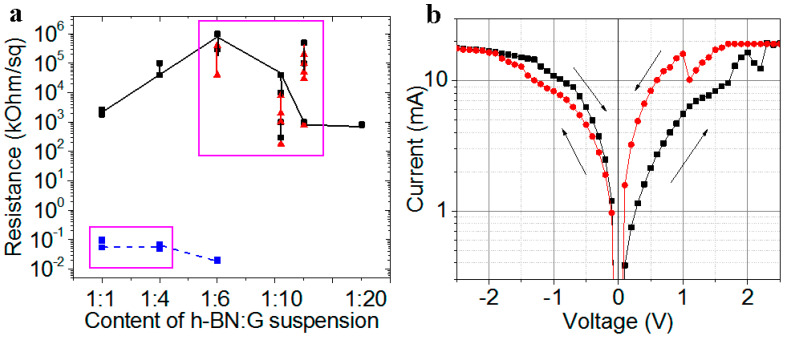
(**a**) Sheet resistance and (**b**) the current-voltage characteristic for the h-BN:G composite and the h-BN:G:PEDOT:PSS composite (content 1:1) with the ethylene glycol additive. G-1 was used in both composites. Rectangles mark the points for which hysteresis was observed on the current-voltage characteristics. The black and red curves correspond to different voltage sweeping directions for composite without the ethylene glycol. Blue points correspond to composite with the ethylene glycol additive.

**Figure 7 nanomaterials-12-01703-f007:**
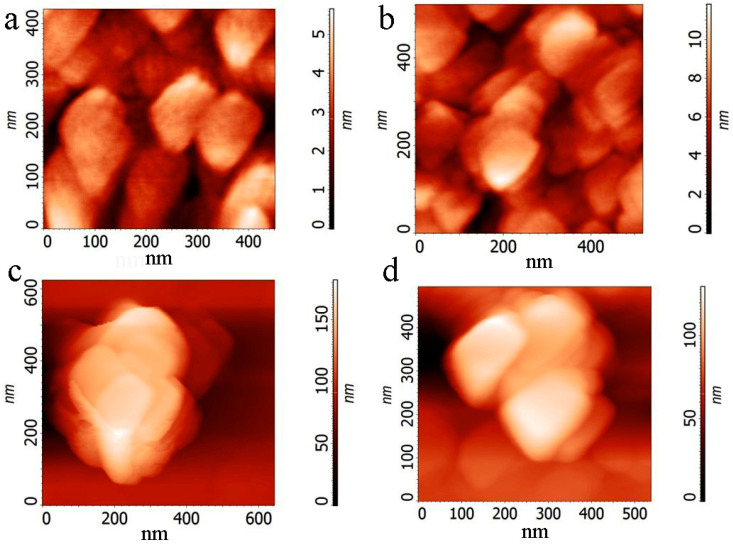
AFM images of the printed films and individual clusters formed from composite suspension h-BN:G:PEDOT:PSS (10^−3^ mass%) with the addition of ethylene glycol (30 vol.%) for h-BN:G content 1:1. (**a**,**c**) G-1, (**b**,**d**) G-2.

**Figure 8 nanomaterials-12-01703-f008:**
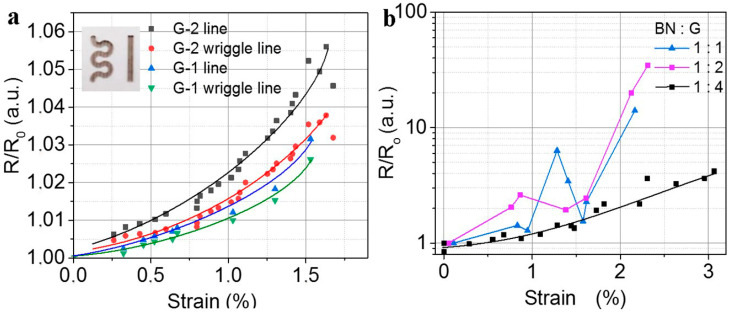
(**a**) Normalized resistance of the printed structures h-BN:G-1 and h-BN:G-2 with the relation of 1:1 in two shapes (straight line and wriggle line, see inserts in (**a**)). (**b**) The normalized resistance of the different composite films created by drops on the PET from G-1. The composite content is given as a parameter.

**Table 1 nanomaterials-12-01703-t001:** Sheet resistance of the 2D printed layers from the h-BN:G:PEDOT; PSS composite suspension with h-BN:G relation 1:1 for the structures on SiO_2_/Si and PET substrates. The number of the printed passes was 30 and 60, respectively. The PEDOT:PSS additive was 10^−3^ mass%, and the ethylene glycol additive was 30 vol.%.

Samples	R, kΩ/, 30 Passes	R, kΩ/, 60 Passes
G-1 on SiO_2_/Si	6.5	2.0
G-1 on PET	9.8	3.5
G-2 on SiO_2_/Si	12.1	5.2
G-2 on PET	27.6	-

## Data Availability

Not applicable.

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
