# Peer review of "Graphene: Hexagonal Boron Nitride Composite Films with Low-Resistance for Flexible Electronics"

_nanomaterials, 2022, doi:10.3390/nano12101703_

Round 1

Reviewer 1 Report

This paper presents the formation of H-BN : graphene-based high conductive composite ink for printing and flexible electronics. Findings have merits in obtaining flexible electrodes using printing techniques for various electronic device applications. This paper could be published after addressing the following questions.

  1. Provide a full name in its first occurrence (h-BN)
  2. What kind of 2D printing technique is suitable for the prepared highly conductive composite inks?
  3. In pages 2 and 3, there are typos in the experimental section.
  4. It would be very useful to mention the time span of the produced ink. How fast the graphene flakes forms cluster in the water? In general, dense clusters affect the printing by clogging the nozzle. Therefore, the Dense cluster formation in BN: G and BN:G:PEDOT: PSS is not suitable for the printing technique. The added videos are not supporting the suitability of the ink for printing.
  5. In figure 6a and 6b, the legend representing black and red lines are missing.
  6. In page 6, it is mentioned that the resistance value of h-BN:G:PEDOT:PSS composite film is in 60-70 Ohms/sq. However, the resistance value in figure 6 is in kOhm/sq. clarify? Verify the unit of resistance?
  7. Explain the reason behind the fluctuation in resistance with an increase in the PEDOT: PSS content (Figure 3) and h-BN: G suspension in Figure 6. Based on the above results, the device-to-device variation seems high. Provide the standard deviation among the similar samples.

Reviewer 2 Report

In this work, the author prepared graphene/h-BN composite with the plasma-chemical approach. The method is interesting. However, the organization of the paper is terrible, especially in language and the data discussion, which prevent the reader from understanding the paper. Major revision should be made. Below are some comments:

  1. The title refer to the high conductive composite films. However, we cannot find the direct data comparison of the conductivity from the paper. the sheet resistance of the film should be transformed to conductivity.
  2. Introduction is bad organized. Specific scientific issue was not provided clearly, which may be due to the poor language. The author should revise the introduction seriously.
  3. “The widely used method of mechanical transfer of individual 2D crystals provides the quality of fabricated heterostructures and interfaces, but it is not scalable.”, related references showed be provided.
  4. In introduction, the author mentioned the composite film shows negative differential resistance or memristor characteristics. What is the reason for the characteristic?
  5. Fig. 1a, we cannot see that it’s monolayer graphene. How the author concluded the 1-3 monolayer flakes?
  6. The preparation of graphene films need to be provided. As far as I know, it is very difficult to prepare large graphene films by solution drying.
  7. Fig. 1c cannot prove the nanoscale of BN particle. It looks like the aggregation.
  8. The ink preparation procedure should be provided, especially the mass ratio or volume ratio of different components. Parameters for ethylene glycol should be given, such as molecular weight.
  9. Why glycol can improve ink viscosity, please provide relevant characterization.
  10. The preparation process of graphene-BN-PEDOT:PSS composite is needed.
  11. In Fig. 5, this is the aggregate, it is general. Generally speaking, this aggregate is harmful to the composite material, both mechanical and electrical properties. Therefore, the author did not show us the advantages of thick plates.
  12. Fig.8b, why do blue and pink curves have big fluctuations?
  13. More references to flexible electronics based on composite materials should be supplemented. Such as: Nature Electronics, 2020, 3(9): 563-570; Nature nanotechnology, 2011, 6(12): 788-792.; Angewandte Chemie International Edition, 2017, 129(30): 8921-8926.;

Reviewer 3 Report

This paper presents a high conductive composite film based on H-BN : graphene. It is impressive the authors suggest adequate insight to the effect of the ratio of the components in the composite. However, the experiments are not sufficient with respect to the insufficiency of the number of samples. Also, there should be additional explanation and data for this work’s reliability. I strongly recommend the authors supplement the followings.

1 The reviewer found several sentences are not well organized. Please check the grammar and revise the sentence appropriately so that readers can understand easily. For example, line 97 in page 2, line 125 in page 3, line 247 in page 9.

2 How many samples have been implemented for the experiments? Especially for the mechanical-electrical property tests in Fig. 8, authors should implement with at least 3 samples. Since the performance of sensors with piezoresistive sensing mechanisms varies a lot with samples, it is necessary to confirm if the results came from the improvement of fabrication or randomly good sample.

3 The authors have not presented any comparison with results in the literature to indicate the significance of their results.

4 Reviewers are confused that the graphene flakes after annealing treatment show better conductivity than one without annealing, why the graphene flakes used for composites were not subjected to annealing.

5 In Figure 4, different ratios of the components in the composite lead to different composite appearances. In order to facilitate comparison, it is recommended to unify the format and scale of each small image in Figure 4.

6 Please explain the reason for the hysteresis in the current-voltage characteristic in Figure 6b.

Round 2

Reviewer 1 Report

The manuscript is look good for publication.

Reviewer 2 Report

I have no comments.  Just a minor suggestion.  It would be better to use "with low-sheet resistance" to substitute the original phrase of "with low-resistance".

Reviewer 3 Report

None